# Prebiotic Effect of Berberine and Curcumin Is Associated with the Improvement of Obesity in Mice

**DOI:** 10.3390/nu13051436

**Published:** 2021-04-24

**Authors:** Audrey M. Neyrinck, Cándido Robles Sánchez, Julie Rodriguez, Patrice D. Cani, Laure B. Bindels, Nathalie M. Delzenne

**Affiliations:** 1Metabolism and Nutrition Research Group, Louvain Drug Research Institute, Université Catholique de Louvain, UCLouvain, B-1200 Brussels, Belgium; audrey.neyrinck@uclouvain.be (A.M.N.); candido.robles@uclouvain.be (C.R.S.); j.rodriguez@uclouvain.be (J.R.); patrice.cani@uclouvain.be (P.D.C.); laure.bindels@uclouvain.be (L.B.B.); 2Walloon Excellence in Life Sciences and BIOtechnology (WELBIO), Louvain Drug Research Institute, UCLouvain, B-1200 Brussels, Belgium

**Keywords:** berberine, curcumin, prebiotic, gut barrier, liver injury, obesity, inflammation

## Abstract

Berberine and curcumin, used as food additives or food supplements, possess interesting anti-inflammatory and antioxidant properties. We tested the potential protective effect of both phytochemicals in genetically obese mice and we determined whether these effects can be related to the modulation of gut functions and microbiota. *Ob/ob* mice were fed a standard diet supplemented with or without 0.1% berberine and/or 0.3% curcumin for 4 weeks. By using targeted qPCR, we found that cecal content of *Bifidobacterium* spp. and *Akkermansia* spp. increased mainly upon berberine supplementation. Genes involved in innate immunity (*Pla2g2a*), mucus production (*Muc2*) and satietogenic peptide production (*Gcg* and *Pyy)* were upregulated in the colon of mice treated with both phytochemicals. Berberine supplementation alone reduced food intake, body weight gain, hypertriglyceridemia and hepatic inflammatory and oxidative stress markers, thus lessening hepatic injury. The increase in *Bifidobacterium* spp. and *Akkermansia* spp. was correlated with the improvement of gut barrier function and with the improvement of hepatic inflammatory and oxidative stresses in obese mice. These data support the fact that non-carbohydrate phytochemicals may modulate the gut microbiota in obesity and related gut and hepatic alterations.

## 1. Introduction

Numerous studies in animal models but also in humans describe alterations of the gut microbial ecosystem (i.e., dysbiosis) associated with obesity and hallmarks of metabolic syndrome, including metabolic-associated fatty liver disease (MAFLD) [1,2,3]. The manipulation of the gut microbiota by the diet appears to be an innovative therapeutic tool to prevent or control obesity and related metabolic disorders [4]. In particular, it has been reported that bacteria such as *Bifidobacterium*, *Lactobacillus* and *Akkermansia muciniphila* may play a crucial role as anti-obesity bacteria in animal models and humans [1,5,6,7,8]. The impact of nutrients that change the gut microbiota composition, or that are metabolized by the gut bacteria into bioactive metabolites, is already described in different animal models of obesity. Interesting strategies to manage weight control are nutritional interventions with dietary fiber [1,9,10]. In particular, prebiotics are defined initially as “a non-digestible food ingredient that beneficially affects the host by selectively stimulating the growth and/or activity of one or a limited number of bacteria in the colon, and thus improves host health” [11]. Inuline-type fructans (fructo-oligosaccharides and inulin) and galactans (galacto-oligosaccharides) are considered key prebiotics through their effect on the enrichment of *Lactobacillus* spp. and/or *Bifidobacterium* spp. [12]. This definition has evolved over time, with the latest consensus being that a prebiotic is a “substrate that is selectively utilized by host microorganisms conferring a health benefit”, thus expanding the concept to non-carbohydrate substances [12]. Phytochemicals, mostly plant polyphenols, have been proposed as potential prebiotics, but more studies are needed to elaborate how their interaction with the gut microbiota influences health effects [12]. It is estimated that 90–95% of dietary polyphenols are not absorbed in the small intestine and, therefore, reach the colon, where they undergo significant biotransformation by the colonic microbiota, often through deglycation and hydrolysis by microbial enzymes, into metabolites that can be absorbed [13]. In fact, growing evidence indicates that the health benefits associated with polyphenol consumption depend more on their microbial metabolism to bioactive molecules than on the parent compounds themselves [13]. However, there are few data on their prebiotic effect (improvement of health with concomitant interaction with the gut microbiota) as compared to a large number of studies performed with fructo-oligosaccharides, galacto-oligosaccharides or inulin-type fructans.

Initially, polyphenols and other phytochemical compounds were extensively studied because of their strong anti-inflammatory and antioxidant properties [14]. Furthermore, preclinical and clinical studies suggest that polyphenols are able to exert antimicrobial activities against pathogenic gut bacteria [15]. Several polyphenols, such as tannins, flavonoids, phenolic acids, lignans and stilbenes, alone or in combination, are known to exhibit antibacterial activity against Gram-positive bacteria. Their mechanism of action is quite diverse, targeting biofilm formation, lipid membrane, membrane receptors, ion channels and bacterial metabolites [16]. More recently, they have attracted the interest of the research community because of their potential role in reducing obesity [17]. Interesting reviews described the effects of plant polyphenols on gut microbiota and/or obesity in preclinical studies [14,18]. Among these, curcumin is often studied. The dried ground rhizome of *Curcuma longa* is a popular dietary spice in Asia, as used in curry [19]. It is also an integral part of the Indian traditional medicine called Ayurveda. The polyphenol curcumin represents 2–8% of most curcuma preparations and is generally regarded as its most active component, having potent antioxidant, anti-inflammatory and anticarcinogenic properties [20,21]. Curcumin-based oral supplements have been proposed in the treatment of various medical inflammatory conditions, such as effluvium telogen [22]. In addition, curcumin has been shown to have protective effects on the liver against fat accumulation, mainly in animal models of high-fat (HF) diet-induced obesity [23]. However, the exact mechanisms by which curcumin reduces liver fat accumulation and alleviates hepatic steatosis are not fully understood. Curcumin also prevents HF diet-induced insulin resistance and obesity by lessening hepatic lipogenesis and inflammation in adipocytes [24]. A few studies demonstrated that dietary curcumin significantly improves obesity-associated inflammation and diabetes in a genetic model of obesity *ob/ob* mice, but the concentration used was very high (1 to 3%) [25,26]. In addition, the potential involvement of gut microbiota modulation in curcumin effects has not been demonstrated in these studies.

Berberine is another interesting prebiotic candidate. It is a natural isoquinoline-type alkaloid originally isolated from *Coptis chinensis*, with a long history of Chinese medicinal application [27]. Berberine suppresses proinflammatory responses by inhibiting mitogen-activated protein kinase signaling and cellular reactive oxygen species production [28]. Based on clinical and preclinical studies, findings support the hypothesis that berberine displays beneficial effects for treating obesity and suggest that the mechanism of action implies AMP-activated protein kinase [29,30].

A synergistic antimicrobial activity of berberine and curcumin has already been demonstrated against methicillin-resistant *S. aureus* (MRSA) [31]. In the present study, we explored the prebiotic potential of both phytochemicals, administrated alone or in combination, in a model of *ob/ob* mice. We targeted by quantitative PCR specific bacteria such as *Lactobacillus* spp., *Bifidobacterium* spp. and *Akkermansia* spp., the well-known bacteria often implicated in the prebiotic effects of nutrients in the context of obesity [5,6,7,8], as well as *Bacteroides* spp. as a good representative dominant Gram-negative bacterium in the gut microbiota [32]. The *ob/ob* genetic model offers the advantage of inducing obesity without using a HF diet, which can indirectly promote the absorption of lipophilic compounds—such as berberine and curcumin—thereby interfering with their potential interaction in the gut microbiota.

## 2. Materials and Methods

### 2.1. Animals and Diet Intervention

Thirty-six B6.V-Lep *ob/ob* JRj 6-week-old male mice (Janvier Labs, Saint Berthevin, France) were housed in groups of 3 per cage in specific pathogen-free and controlled environment (12-h daylight cycle) with free access to food and water. Mice were acclimated with a standard diet (AIN93M pellets, D10012Mi, Research Diet, New Brunswick, NJ, USA) for one week and then assigned into 4 groups of 9 mice: mice fed a standard diet (cc-/bb-, *n* = 9), mice fed a standard diet supplemented with 0.1% berberine (cc-/bb+, *n* = 9), mice fed a standard diet supplemented with 0.3% curcumin (cc+/bb-, *n* = 9) and mice fed 0.1% berberine and 0.3% curcumin (cc+/bb+, *n* = 9). Eleonor (Waterloo, Belgium) supplied the berberine and curcumin extracts. The powdered form of the AIN-93M diet (D10012Mmi, Research Diet, New Brunswick, NJ, USA) was mixed with the compounds, following guidelines on Good Pharmacy Practice (method of successive addition, www.fip.org (accessed on 8 April 2021)). Sterile water was added in the different mixtures (300 mL water/kg diet) to prepare pastes that were then divided into pellets (2 × 2 × 0.5 cm) and dried at room temperature. Food intake (for 3 mice per cage, *n* = 3) and body weight were recorded twice a week. After 4 weeks of dietary treatment, 6-h-fasted mice were anaesthetized (isoflurane gas, Abbot, Ottignies, Belgium). Blood glucose was determined with a glucose meter (Roche Diagnostic, Meylan, France) on 3.5 µL blood collected from the tip of the tail vein. Blood samples were harvested for further analysis. Mice were necropsied after cervical dislocation. Adipose tissues (epididymal, visceral, brown), liver, cecal content and tissue were weighted. Cecal content and tissue, colon and liver were frozen in liquid nitrogen and stored at −80 °C. Animal experiments complied with the commonly accepted ‘3Rs’ and were approved and performed in accordance with the guidelines of the local ethics committee for animal care of the Health Sector of the Université catholique de Louvain under the specific agreement number 2017/UCL/MD/005 (date of approval: 15/05/2017). Housing conditions were as specified by the Belgian Law of 29 May 2013, on the protection of laboratory animals (Agreement LA 1230314). All experiments were performed in strict accordance with relevant guidelines and regulations for the care and use of animals and in accordance with the EU directive.

### 2.2. Gut Bacteria Analysis

Genomic DNA was extracted from the cecal content using a QIAamp DNA Stool Mini Kit (Qiagen, Hilden, Germany) according to the manufacturer’s instructions, including a bead-beating step. Quantification of total bacteria, *Bacteroides* spp., *Bifidobacterium* spp., *Akkermansia* spp. and *Lactobacillus* spp., was performed by qPCR, as described earlier [33].

### 2.3. Biochemical Analysis

Plasma triglycerides, cholesterol and non-esterified fatty acid (NEFA) concentrations were measured using kits coupling enzymatic reactions and spectrophotometric detections (SpectraMax i3x, Molecular Devices, UK) of end products (Diasys Diagnostic and Systems, Holzheim, Germany). Plasma insulin concentration was determined using ELISA kit (Mercodia, Upssala, Sweden). Triglycerides and cholesterol were measured in the liver tissue according to the Folch method [34] and using kits (Diasys Diagnostic and System, Holzheim, Germany). Alanine aminotransferase (ALAT) and aspartate aminotransferase (ASAT) levels were measured in the plasma as markers of liver damage using commercial kits according to the manufacturer’s instructions (Diasys Diagnostic and Systems, Holzheim, Germany).

### 2.4. Tissue mRNA Analyses

Total RNA was extracted from tissues using TriPure Isolation Reagent (Roche Diagnostics, Belgium). Complementary DNA was prepared by reverse transcription of 1 µg of total RNA using the Reverse Transcription System (Promega, Madison, WI, USA). Real-time PCR was performed with a QuantStudio 3 Real-Time PCR System (Applied Biosystems, Den Ijssel, The Netherlands) using MasterMix GoTaq (Promega, WI, USA). Data were analyzed according to the 2-ΔΔCT method. The purity of the amplified products was verified by analyzing the melting curve obtained at the end of the amplification step. The ribosomal protein L19 (Rpl19) gene was used as a reference gene. Primer sequences are given in Appendix A.

### 2.5. Statistical Analysis

Results are presented as means with their standard errors. Statistical analysis was performed by two-way analysis of variance (ANOVA) followed by post-hoc Tukey’s multiple comparison tests using Graphpad Prism software version 8 (San Diego, USA; www.graphpad.com (accessed on 8 April 2021)). The results were considered statistically significant when *p* value was < 0.05.

## 3. Results

### 3.1. Supplementation with Berberine Reduced Body Weight Gain and Food Intake of Ob/Ob Mice, Independently of Curcumin

Body weight of *ob/ob* mice increased significantly with time, regardless of the dietary treatment (Appendix A). Body weight gain of obese mice was lower upon berberine supplementation (Appendix A). This effect was linked to a lower food intake upon berberine intake, independently of curcumin intake (Table 1, Appendix A). However, we did not find any significant effect of berberine on adiposity or liver weight. We observed a higher proportion of brown adipose tissue upon curcumin supplementation. The cecal content was significantly higher in the berberine groups. However, this effect was not accompanied by an increase in the cecal tissue.

### 3.2. Supplementation with Berberine or Curcumin Has Prebiotic Potential in Ob/Ob Mice

Berberine supplementation significantly decreased total bacteria in the cecal content, independently of the presence of curcumin in the diet (Figure 1a). The decrease in lactobacilli after berberine intake partially explained this effect (Figure 1b). *Bacteroides* spp. were not affected by the dietary treatments (Figure 1c). In contrast, *Bifidobacterium* spp. and *Akkermansia* spp. were increased by the berberine treatment (Figure 1d,e). Interestingly, the number of bifidobacteria was also significantly higher upon curcumin supplementation (* *p* < 0.05, two-way ANOVA).

### 3.3. Supplementation with Berberine Upregulates the Expression of Host Peptides in the Colon of Ob/Ob Mice

We analyzed the mRNA levels of two colonic precursors of satietogenic peptides (*Gcg* coding for glucagon-like peptides GLP-1 and GLP-2; *Pyy* coding for peptide YY), of four important inflammatory markers (*Tnf, Il1b, Il6* and *Ifng*), as well as of peptides/proteins able to influence gut barrier function, such as *Muc2* involved in mucin production, and tight junction proteins (*Tjp1* and *Ocln*) (Figure 2, Appendix A). Furthermore, we analyzed colonic expression of antimicrobial peptides, which can explain how nutrients modulate the gut ecosystem [35,36]. We measured the expression of antimicrobial peptides produced by enterocytes and Paneth cells in the colon: regenerating islet-derived 3-gamma (*Reg3g*), phospholipase A2g2 (*Pla2g2a*) and lysozyme C (*Lyz*) (Figure 2d, Appendix A). We observed decreased mRNA contents of *Muc2*, *Pla2g2a* and *Gcg* in the curcumin+/berberine- group. Interestingly, berberine increased the *Pyy* mRNA level and counteracted the decreased expression in genes induced by curcumin. In contrast, the expression of Reg3γ was downregulated upon berberine supplementation, mainly in the absence of curcumin in the diet, whereas the *Lyz* mRNA level was unchanged (Appendix A). Although occludin was also downregulated by curcumin supplementation, as observed above for other peptides, berberine intake did not influence its expression (Appendix A). The expression of a tight junction protein (*Tjp1*) and four important inflammatory markers in the colon of obese mice was not affected by dietary treatments (Appendix A).

### 3.4. Supplementation with Berberine Does Not Change Plasma Fasting Glucose and Insulin Levels but Decreases Triglyceridemia in Ob/Ob Mice

Berberine alone or in combination with curcumin in the diet had no effect on fasting glycemia or fasting insulinemia in obese mice (Figure 3a,b). Of particular interest, berberine decreased plasma triglycerides of obese mice, independently of the presence of curcumin in the diet (Figure 3c). No effect was observed on cholesterolemia (Figure 3d).

### 3.5. Supplementation with Berberine Decreases Oxidative and Inflammatory Stresses in the Liver of Ob/Ob Mice

In accordance with the histological analysis (data not shown), the hepatic content of lipids (triglycerides and cholesterol) was not affected by the dietary treatments (Table 2). Although we observed the lowest triglyceride content in the liver of mice treated with curcumin only, this decrease was not significant and was even blunted by the presence of berberine in the diet. Importantly, berberine supplementation decreased the levels of ALAT, reflecting a lower hepatic injury in obese mice (Table 2). In line with this result, berberine treatment decreased markers of activated macrophages (*Itgax* coding for CD11c) and of oxidative stress (*Nox1* coding for NADPH-oxidase). However, these results were not accompanied by lower expression of TNFα, IL6, monocyte chemoattractant protein 1 (MCP1 or *Ccl2*) and toll-like receptor 4 (TLR4) (Table 2). Of note, the antioxidant effect of berberine was observed only in the presence of curcumin in the diet.

### 3.6. Increase in Beneficial Gut Bacteria upon Berberine or Curcumin Supplementation Correlates with Favorable Changes in Host Parameters in Ob/Ob Mice

We performed Spearman correlation analysis on data obtained from all mice (*n* = 36) in order to evaluate the potential links between the significant increase in specific gut bacteria induced by both phytochemicals (i.e., *Bifidobacterium* spp. and *Akkermansia* spp.) and all host parameters significantly affected (increased or decreased) mainly by the berberine treatment (Table 3). We observed significant negative correlations between body weight gain, triglyceridemia, parameters related to liver inflammation and oxidative stress with both *Bifidobacterium* spp. and *Akkermansia* spp. Moreover, colonic production of the satietogenic peptide YY was correlated with both beneficial *Bifidobacterium* spp. and *Akkermansia* spp.

## 4. Discussion

The gut microbiota is increasingly considered a symbiotic partner of health. Several data obtained in animal studies but also in humans suggest that the activity of the gut microbiota is a key factor to take into consideration when assessing the metabolic disorders related to obesity, such as insulin resistance, inflammation, dyslipidemia and MAFLD [1,2,3,37]. Emerging evidence has supported the relevance of some bacteria as hallmarks of obesity. The number of bifidobacteria has been inversely correlated with many disorders occurring upon obesity, such as adiposity, glucose intolerance and endotoxemia, notably in *ob/ob* mice models [38,39]. Intervention studies with bifidobacteria as probiotics, or with prebiotics known to stimulate bifidobacteria growth, such as inulin and fructo-oligosaccharides, were tested as promising approaches to counteract some components of obesity and related pathologies [40]. *Akkermansia muciniphila* is attracting increasing interest for its health benefits [41]. In particular, Zhang et al. found a decreased abundance of *Akkermansia muciniphila* in diabetic and glucose-intolerant patients; this observation has been reported in several studies of obese individuals [42]. Among gut bacteria, *Akkermansia* is a mucus-colonizing bacterium involved in gut barrier function [43]. Treatment of rodents with *A. muciniphila* reduces obesity and related disorders, such as gut permeability, insulin resistance, glucose intolerance and steatosis. The hypothesis of specific changes in the abundance of bifidobacteria and *Akkermansia* in obesity is supported by several studies performed in mice as well as in humans [1,6,44]. To date, dietary fibers have been proposed as the key nutrients able to promote such interesting bacteria. In the present study, we extend the concept of prebiotics to non-carbohydrate phytochemicals. Previous studies using untargeted approaches to analyze the gut microbiota composition revealed that the main changes concerned with fostering the growth of *Akkermansia* spp., *Bacteroides* spp. and/or *Lactobacillus* spp. abundance in several animal models of metabolic disorders (HF diet-induced atherosclerosis in Apoe-/- mice, *db/db* and KKAy diabetic mice) justified the targeting of those species when berberine impact on gut microbiota was investigated [45,46,47]. Therefore, we analyzed the gut microbiota composition using quantitative PCR targeting *Akkermansia* spp., *Bifidobacterium* spp., *Lactobacillus* spp. and *Bacteroides* spp., this methodology being more specific and quantitative than untargeted analyses (such as 16S rRNA gene sequencing). We have observed that the co-administration of curcumin and berberine exerted additive effects (i.e., no significant interaction in ANOVA as defined by Piggott et al., [48]) on *Bifidobacterium* spp. level. In addition, an important increase in *Akkermansia* was only observed after berberine intake, in line with other studies showing berberine’s impact on this interesting bacterium [45,46,47,49]. The limitation is that we cannot exclude that other bacteria than the ones measured have been affected by the dietary treatments and may contribute to the effects demonstrated in the present study. For example, a detailed analysis of the gut microbiota in diabetic *db/db* mice using 16 s rDNA high-throughput sequencing indicated that the proportions of *Butyricimonas, Lactobacillus, Coprococcus* and *Ruminococcus* increased in addition to *Akkermansia*, while the proportions of *Prevotella* and *Proteus* were reduced in the berberine-treated group [46]. Interestingly, in this last study, *Bacteroides, Parabacteroides* and *Proteobacteria* were positively correlated with weight or blood glucose, whereas *Akkermansia, Sutterella, Clostridium, Butyricimonas* and *Bifidobacterium* were negatively associated with these factors. Other bacteria belonging to *Lachnospiraceae* and *Desulfovibrionaceae* may also decrease upon berberine treatment in diabetic mice [45]. Furthermore, the abundance of *Terrisporobacter* and *Helicobacter* has been shown to increase while that of *Pseudoflavonifractor, Mucisirillum, Alistipes, Ruminiclostridium* and *Lachnoclostridium* decreased after berberine treatment in an animal model of alcoholic liver disease, even if *A. muciniphila* was the one that was most remarkably influenced by berberine administration [49]. By which mechanisms would berberine intake affect the composition of the gut microbiota? Multiple factors may contribute to changes in the intestinal microbiota, one of them being the interaction with the innate immune system of the gut. Antimicrobial molecules belonging to the innate immune system are secreted from epithelial cells. In the present study, the mRNA level of *Pla2g2a* in the colon increased, whereas that of *Reg3g* decreased upon berberine supplementation. Strikingly, although we found a decrease in *Reg3g* in this study, several previous studies have shown that direct supplementation with *Akkermansia muciniphila* increases the expression of *Reg3g* mRNA in the colon, and the higher abundance of *Akkermansia* observed upon prebiotic feeding was correlated with *Reg3g* [36,43,50,51]. This suggests that the host contributes to the modification of the gut microbiota induced by berberine extract supplementation, by modulating the production of such antimicrobial peptides [36,52]. However, whether *Akkermansia* directly influences the expression of *Reg3g* or *vice versa* remains to be investigated. In addition, we cannot exclude the direct antibacterial activity of curcumin and berberine since synergistic antimicrobial activity has already been demonstrated in vitro [31]; this mechanism can contribute, for example, to the lower proportion of *Lactobacillus* species observed in the cecal content mainly occurring in the curcumin+/berberin+ group. Altogether, these results suggest a prebiotic capacity of both chemicals in a genetic model of obesity.

Prebiotics such as fructans are able to prevent the overexpression of several host genes related to adiposity and inflammation, whereas they can increase the expression of colonic genes involved in the production of PYY, GLP-1 and GLP-2 in the portal vein, notably in *ob/ob* mice in a 6-h-fasting state [3,38,53]. PYY and GLP-1 have a role in the control of appetite and glucose homeostasis, whereas GLP-2 contributes to the improvement of gut barrier function, namely through the higher expression of tight junction proteins. In the present study, we did not observe any increase in the tight junction proteins (TJP1, i.e., zonula occludens 1) or occludin mRNA levels under curcumin or berberine supplementation. However, we cannot exclude a modification by dietary treatments at the protein level, as already demonstrated with berberine supplementation by immunochemistry for occludin in the intestinal epithelium of obese rats due to HF feeding [54]. The gut barrier that controls the passage of gut-derived lipopolysaccharides (LPS) into the circulation is regulated by the intestinal inner mucus layer in addition to the presence of tight junctions. It was shown that *A. muciniphila* increases the mucus layer and its level is positively correlated with the number of mucin-producing goblet cells [43]. Here, we observed higher colonic expression of *Muc-2* involved in mucin production in the curcumin+/berberine+ treated mice, suggesting an effect on the mucus layer, as already demonstrated for berberine, together with a higher abundance of *Akkermansia* spp. in HF diet-induced atherosclerosis Apoe-/- mice [47]. Importantly, we have shown higher colonic mRNA content of precursors of satietogenic peptides (*Gcg* and *Pyy* coding for GLP-1 and PYY, respectively) after berberine intake. These parameters have been measured after 6 h of fasting in accordance with previous protocols [3,38,53]. We do not know if similar effects would be relevant in the fed state; this could be one limitation of the outcome. The effect of berberine on preproglucagon and PYY expression was conditioned to the presence of curcumin in the diet. Of particular interest, berberine was reported to restore aberrant levels of gut hormones such as GLP-1, GLP-2 and PYY in the portal plasma of rats fed a HF diet [55]. The regulation of gut peptide production occurring in the colon of obese mice fed a diet enriched with berberine might be involved in the lower food intake observed in the present study, a mechanism that could participate in the lower body weight gain, as already described for inulin-type fructans [3,53].

We have previously highlighted that plasma hypertriglyceridemia was dramatically reduced in *ob/ob* mice treated with prebiotics for 5 weeks [39]. Here, this metabolic alteration was improved upon berberine supplementation after 4 weeks of treatment, independently of curcumin intake. However, 4 weeks of treatment with berberine or curcumin, at a concentration in the diet of 0.1% and 0.3%, respectively, was insufficient to improve glucose homeostasis in obese mice in view of the unchanged levels of glycemia or insulinemia.

Concurrently with the increase in obesity, non-alcoholic fatty liver disease (NAFLD, recently defined as MAFLD) is becoming the most important cause of chronic liver disease, with a global prevalence of 24%. However, no pharmacological therapy is currently available [56]. Diet plays an essential role in the development of MAFLD, but growing evidence suggests that the gut microbiota also has its part to play in the occurrence and evolution of this disease, notably through the production of specific bioactive metabolites [57,58]. The present work highlighted the hepatoprotective properties of supplementation with berberine. Berberine decreased oxidative (lower expression of *Nox1* coding for NADPH-oxidase) and inflammatory events (lower expression of CD11c, a marker of activated macrophages) in the liver, leading to a lower hepatic injury (decrease in ALAT level in the plasma) in a genetic model of obesity and associated NAFLD, without improving steatosis and lipid homeostasis. It has also been reported that berberine supplementation may downregulate the levels of ALAT in rats fed a HF diet inducing obesity, indicating the restoration of liver function [54]. In this last study, and in line with our results, it was shown that berberine provided significant protection in NAFLD through ameliorating intestinal barrier function.

Finally, we highlighted significant correlations between *Bifidobacterium* and *Akkermansia* levels and the improvement of most metabolic alterations, including body weight gain, triglyceridemia, satietogenic peptide YY expression, liver inflammation and oxidative stress.

In conclusion, these data supported the fact that non-carbohydrate phytochemicals may modulate the gut microbiota in obesity and related gut and hepatic alterations. This study thus strengthens the beneficial aspects of phytochemicals as prebiotic nutrients on body weight gain and food intake, an effect involving the production of satietogenic peptides in the colon. Moreover, these new prebiotic candidates improve obesity-related metabolic disorders such as hypertriglyceridemia, liver inflammation and hepatic oxidative injury when obesity is already established. It would be interesting to study the role of key metabolites produced by the gut microbiota from curcumin and berberine in order to propose the molecular mechanism(s) behind the systemic effects of these phytochemicals. Metabolomic approaches could be useful for this purpose. In addition, appropriate placebo-controlled intervention studies in overweight or obese individuals would also constitute a useful perspective to elaborate the proof of concept of their relevance in the management of obesity-related diseases in humans.

## Figures and Tables

**Figure 1 nutrients-13-01436-f001:**
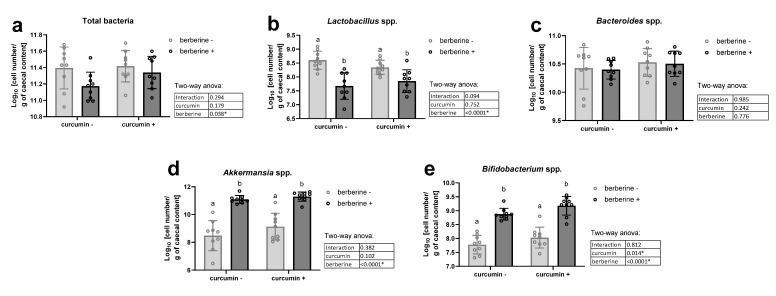
Bacteria in the cecal content. *Ob/ob* mice were fed a standard diet supplemented with or without berberine or curcumin for 4 weeks (*n* = 9 for each group). qPCR analysis of total bacteria (**a**), *Lactobacillus* spp. (**b**), *Bacteroides* spp. (**c**), *Akkermansia* spp. (**d**) and *Bifidobacterium* spp. (**e**) in the caecal content. * *p* < 0.05 for berberine or curcumin effect (two-way ANOVA). Data with different superscript letters are significantly different at *p* < 0.05 (Tukey post-hoc test).

**Figure 2 nutrients-13-01436-f002:**
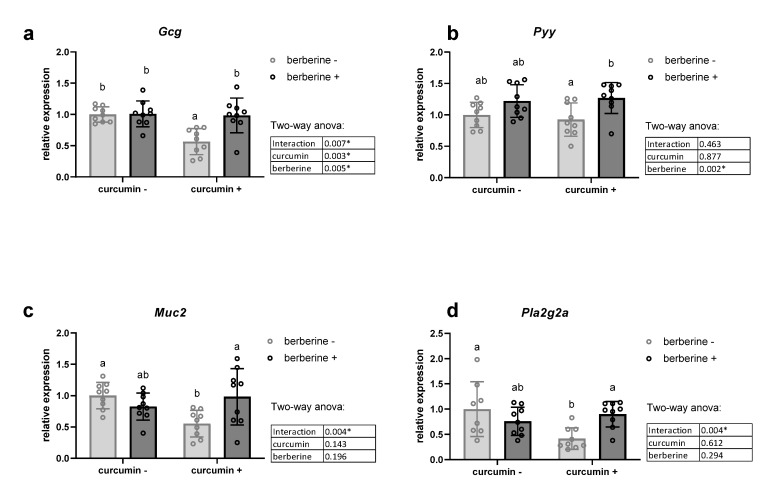
Colonic mRNA level coding for host peptides in the colon. *Gcg* coding for glucagon-like peptides (GLP-1, GLP-2) (**a**), *Pyy* coding for peptide YY (**b**), *Muc2* coding for mucin 2 (**c**) and *Pla2g2a* coding for phospholipase A2 group II (**d**). *Ob/ob* mice were fed a standard diet supplemented with or without berberine or curcumin for 4 weeks (*n* = 9 for each group). * *p* < 0.05 for berberine or curcumin effect (two-way ANOVA). Data with different superscript letters are significantly different at *p* < 0.05 (Tukey post-hoc test).

**Figure 3 nutrients-13-01436-f003:**
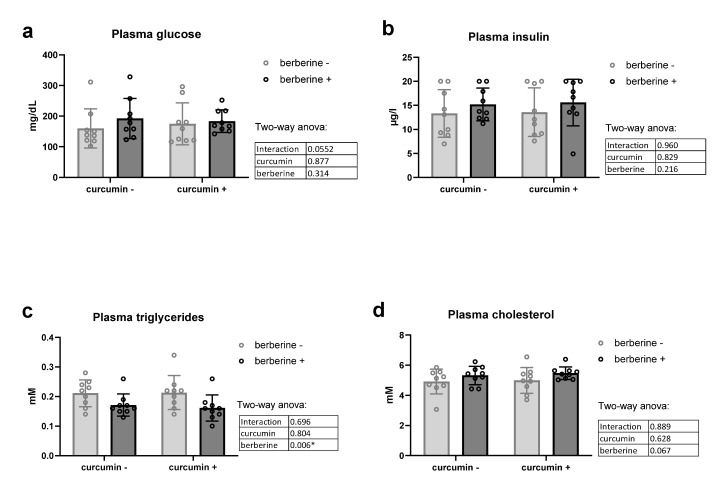
Lipid profile and glucose homeostasis of *ob/ob* mice fed a standard diet supplemented with or without berberine or curcumin for 4 weeks (*n* = 9 for each group). Plasma glucose (**a**), plasma insulin (**b**), plasma triglycerides (**c**), plasma cholesterol (**d**) * *p* < 0.05 for berberine effect (two-way ANOVA).

**Table 1 nutrients-13-01436-t001:** Body weight gain, food intake and organ weights.

	cc-/bb-	cc-/bb+	cc+/bb-	cc+/bb+
Body weight gain (g) *	5.39 ± 0.33 ^a^	3.66 ± 0.29 ^b^	4.76 ± 0.16 ^ab^	3.64 ± 0.40 ^b^
Total food intake (g/cage) *	93.9 ± 2.6 ^a^	82.4 ± 2.2 ^b^	94.1 ± 3.2 ^a^	79.9 ± 0.6 ^b^
Visceral adipose tissue (g)	1.11 ± 0.05	1.07 ± 0.03	1.17 ± 0.04	1.10 ± 0.03
Epididymal adipose tissue (g)	3.21 ± 0.08	3.24 ± 0.14	3.18 ± 0.09	3.05 ± 0.06
Brown adipose tissue (g) #	0.21 ± 0.01	0.19 ± 0.02	0.24 ± 0.01	0.24 ± 0.02
Liver (g)	2.81 ± 0.16	2.89 ± 0.13	3.11 ± 0.11	2.78 ± 0.10
Cecal content (g) *	0.13 ± 0.02	0.16 ± 0.01	0.12 ± 0.01	0.14 ± 0.01
Cecal tissue (g)	0.050 ± 0.002	0.058 ± 0.003	0.054 ± 0.003	0.053 ± 0.003

*Ob/ob* mice were fed a standard diet supplemented with or without berberine (bb) or curcumin (cc) for 4 weeks (*n* = 9 for each group). * *p* < 0.05 for berberine effect and # *p* < 0.05 for curcumin effect (two-way ANOVA). Data with different superscript letters are significantly different at *p* < 0.05 (Tukey post-hoc test).

**Table 2 nutrients-13-01436-t002:** Hepatic parameters related to lipid accumulation, inflammation and hepatotoxicity.

	cc-/bb-	cc-/bb+	cc+/bb-	cc+/bb+
ALAT (U/L) *	150 ± 12	130 ± 6	158 ± 11	137 ± 7
Triglyceride content (nmol/mg tissue) ^§^	188 ± 11	171 ± 10	157 ± 7	182 ± 6
Cholesterol content (nmol/mg tissue)	60 ± 3	55 ± 3	61 ± 4	60 ± 2
*Tnf* (relative expression) ^§^	1.00 ± 0.08	1.92 ± 0.44	2.36 ± 0.61	1.24 ± 0.23
*Il6* (relative expression) ^§^	1.00 ± 0.12	1.70 ± 0.21	1.83 ± 0.38	1.27 ± 0.19
*Ccl2* (relative expression)	1.00 ± 0.11	0.98 ± 0.12	1.22 ± 0.12	1.11 ± 0.27
*Tlr4* (relative expression)	1.00 ± 0.05	1.35 ± 0.26	1.71 ± 0.50	1.69 ± 0.65
*Itgax* (relative expression) *	1.00 ± 0.14	0.81 ± 0.06	1.19 ± 0.16	0.84 ± 0.12
*Nox1* (relative expression) *^§^	1.00 ± 0.07 ^a^	1.01 ± 0.05 ^a^	1.10 ± 0.07 ^a^	0.73 ± 0.07 ^b^

*Ob/ob* mice were fed a standard diet supplemented with or without berberine or curcumin for 4 weeks (*n* = 9 for each group). * *p* < 0.05 for berberine effect, ^§^
*p* < 0.05 for interaction effect (two-way ANOVA). Data with different superscript letters are significantly different at *p* < 0.05 (Tukey post-hoc test). ALAT, alanine aminotransferase.

**Table 3 nutrients-13-01436-t003:** Correlation analysis between increased cecal bacteria and host parameters significantly modulated by berberin or curcumin treatments ^1^.

Spearman r	*Bifidobacterium* spp.	*Akkermansia* spp.
Body weight gain	−0.60 *	−0.53 *
Plasma triglycerides	−0.42 *	−0.41 *
Brown adipose tissue weight	0.05	0.10
ALAT	−0.29	−0.25
Colonic *Muc2*	0.03	0.14
Colonic *Pla2g2a*	0.21	0.29
Colonic *Reg3g*	−0.61	−0.58 *
Colonic *Pyy*	0.39 *	0.42 *
Colonic *Gcg*	0.18	0.29
Liver *Nox1*	−0.37 *	−0.40 *
Liver *Itgax*	−0.32	−0.34 *

^1^ Spearman r coefficients were computed between bacteria significantly increased after berberine and curcumin intake and all host parameters significantly affected by the dietary treatments (*n* = 36, * *p* < 0.05). ALAT, alanine aminotransferase.

## Data Availability

Data described in the manuscript will be made available upon reasonable request from the corresponding author.

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
