# Peer review of "Prebiotic Effect of Berberine and Curcumin Is Associated with the Improvement of Obesity in Mice"

_nutrients, 2021, doi:10.3390/nu13051436_

Round 1
Reviewer 1 Report
An interesting original study on the use of berberine and curcumin supplementation in the diet of mice, improving obesity-related metabolic disorders such as hypertriglyceridemia, liver inflammation, and hepatic oxidative injury. Only minor queries:
A conclusion paragraph, suggesting the future possible developments of this study findings, as well as the new possible studies that could be performed, would be in my opinion a great addition.
Page 2 line 62-63 "Curcumin based oral supplements have been proposed in the treatment of various medical inflammatory conditions, such as effluvium telogen" and cite an article such as: doi: 10.1111/dth.12842.
Thank You
Author Response
We thank the reviewers for the useful comments. We performed several changes in the text and added some data to meet the comments and suggestions of the reviewers. Furthermore, the spelling has been checked as suggested by the reviewers. We have rephrased and highlighted the modified sentences in the version with track changes.

Reviewer 2 Report
In the Introduction the Authors are missing a key point and references about the many known anti-microbial effects of polyphenols and polyphenol-rich foods in explaining some of their benefits. This is important information to provide to the reader regarding the study. The study focuses on changes on a limited selection of microbes via Q-PCR rather than examining the breadth of changes in gut microbes through sequencing methods. This is a deficiency of the study in understanding the full effects of the treatments on the gut microbes. The Discussion should include points indicating that impacts on other microbes are likely, and that this would be uncovered through deeper analysis via sequencing methods. Some literature on microbiota impacts of polyphenols and your supplements should be expanded upon here.
The paper does not provide any background information in the Introduction that helps the reader understand the choice of microbe targets analysed by Q-PCR. The study is about prebiotic potential. Please explain how the targets such as Bifidobacterium and Lactobacillus facilitate that goal. The results section 3.2 ‘Supplementation with berberine or curcumin has prebiotic potential in ob/ob mice’ assumes knowledge about what is regarded as prebiotic potential.
In the methods, please indicate the number of mice per treatment group.
In the methods, please indicate the frequency of feed intake and body weight monitoring.
The information provided in Table 1, methods and results is insufficient in relation to weight gains and intakes. When and over what periods of the experiment were the measures taken? Please provide the average weights of animals when received or at the start of the 4-week treatment.
Please indicate in the text how the standard diet was supplemented. Was the commercial diet crushed and reformulated with the supplements? Were the diets pelleted or in powder form?
I don’t understand section 3.6 and Table 3. The heading indicates correlations performed between host parameters and bacteria significantly modulated by ‘berberin or curcumin treatments’. The text under the table suggests the analysis was done on bacteria increased after ‘berberine and curcumin intake’. This is very unclear. Was the correlation done on data from all treatment groups or selected treatment groups? Please make the explanations clearer.
Please provide the replication number (n=) in legends of tables and figures.
I am particularly concerned about the results of 3.4 Supplementation with berberine does not change the diabetic status but decreases triglyceridemia in ob/ob mice. It is stated that ‘Berberine alone or in combination with curcumin in the diet had no effect on fasting hyperglycemia or fasting hyperinsulinemia of obese mice (Figure 3 AB)’. How have the authors determined that the animals have diabetes? How have the authors made a determination of fasting hyperglycaemia and hyperinsulinaemia? The methods simply indicate that animals were fasted for 6 h prior to being euthanased. There is no indication that blood was collected at the start of the 6h fast or other times during the study. Glucose and insulin measures would need to be made prior to the fast to assess effects of the fast. Without the pre-6h timepoint these results do not have meaning.
The labelling within Figure 3 needs to be modified. The y-axes do not indicate what is being measured. The heading on each figure doesn’t either. Glycemia is not the component being measured, presumably it is glucose. There is a similar issue with each graph within Fig 3.
Another significant concern relates to the impact of a 6h fast on expression of genes related to metabolic activities (oxidative stress) and also the peptides involved in satiety. These would almost certainly be influenced by fasting for 6 h. How can the authors make a true interpretation of the effects of their treatment over 4 wks when the fast will significantly disrupt many parameters measured? What was the purpose of the 6 h fast?
Author Response

(The authors gave the same response as above.)

Round 2
Reviewer 2 Report
The Authors have satisfactorily addressed most of the concerns raised. However, some concerns remain.
While not raised initially, I note that the title is misleading in that it suggests a prebiotic effect is shown to ‘contribute’ to obesity-related outcomes. Its not clear from the data in the study that a direct causal relationship can be made between the changes in the microbes and beneficial outcomes. These appear to only be associations. In fact your concluding paragraph of the paper suggests it would be interesting to understand which microbial metabolites are responsible for effects, implying a lack of clear causality. Please reword the title to account for this discrepancy. Perhaps replace with ‘is associated with ..’ rather than ‘contribute to’. It would also be appropriate to indicate in the title that this outcome is in mice. In light of the lack of clear evidence of a link between the microbes and obesity outcomes I do not believe the final sentence of the Abstract, or similar comments elsewhere, is valid as written (i.e. Those data enlarged the prebiotic concept to non-carbohydrates phytochemicals that are able to improve obesity and related gut and hepatic alterations).
In the revised statements added to the Introduction, the use of ‘Incriminate’ isn’t the right word. Its usually used for blame for something bad. Do you mean implicated?
In your response to point 2, the Authors have indicated ‘The limitation is that we cannot exclude that other bacteria than the ones measured in this study have been affected by the dietary treatments.’ It was suggested in the initial review that a more substantial Discussion of the role of polyphenols, including those related to the study, be made. While briefly stated in the Introduction, there is no follow up of this subject in the Discussion. I think it is more than ‘cannot exclude’ in this study. Indeed, it is likely. It is appropriate to provide a more substantial discussion of this topic of the other microbes that may contribute to your findings. Please provide that in the Discussion.
I continue to have a concern about the effects of fasting on the analyses, particularly the effects on metabolic parameters/gene expressions. The effects of dietary treatments are potentially being modulated by the 6 h fast. This potential should at least be carefully discussed, which it is not. Please do so.
Author Response
Dear reviewer, we performed all required changes in the text. We have rephrased and highlighted the modified sentences in the version with track changes.
